# A Study on the Functional Identification of Overexpressing Winter Wheat Expansin Gene *TaEXPA7-B* in Rice under Salt Stress

**DOI:** 10.3390/ijms25147707

**Published:** 2024-07-14

**Authors:** Xue Wang, Jing Ma, Fumeng He, Linlin Wang, Tong Zhang, Dan Liu, Yongqing Xu, Fenglan Li, Xu Feng

**Affiliations:** College of Life Sciences, Northeast Agricultural University, Harbin 150030, China; wang_x2022@126.com (X.W.); 15184523620@163.com (J.M.); hefumeng@neau.edu.cn (F.H.); 18315407194@163.com (L.W.); 15147138029@139.com (T.Z.); b230902001@neau.edu.cn (D.L.); yuti8221@163.com (Y.X.)

**Keywords:** winter wheat, expansin gene *TaEXPA7-B*, functional identification, rice, salt stress

## Abstract

Expansin is a cell wall relaxant protein that is common in plants and directly or indirectly participates in the whole process of plant root growth, development and morphogenesis. A well-developed root system helps plants to better absorb water and nutrients from the soil while effectively assisting them in resisting osmotic stress, such as salt stress. In this study, we observed and quantified the morphology of the roots of *Arabidopsis* overexpressing the *TaEXPAs* gene obtained by the research group in the early stage of development. We combined the bioinformatics analysis results relating to EXPA genes in five plants and identified *TaEXPA7-B*, a member of the EXPA family closely related to root development in winter wheat. Subcellular localization analysis of the TaEXPA7-B protein showed that it is located in the plant cell wall. In this study, the *TaEXPA7-B* gene was overexpressed in rice. The results showed that plant height, root length and the number of lateral roots of rice overexpressing the *TaEXPA7-B* gene were significantly higher than those of the wild type, and the expression of the *TaEXPA7-B* gene significantly promoted the growth of lateral root primordium and cortical cells. The plants were treated with 250 mM NaCl solution to simulate salt stress. The results showed that the accumulation of osmotic regulators, cell wall-related substances and the antioxidant enzyme activities of the overexpressed plants were higher than those of the wild type, and they had better salt tolerance. This paper discusses the effects of winter wheat expansins in plant root development and salt stress tolerance and provides a theoretical basis and relevant reference for screening high-quality expansin regulating root development and salt stress resistance in winter wheat and its application in crop molecular breeding.

## 1. Introduction

Winter wheat (*Triticum aestivum*) is one of the most important staple crops in the world [1]. The winter wheat region in northern China is one of the most important wheat-producing areas in China, accounting for about 60% of the country’s cultivated area and 80% of the country’s wheat yield [2,3]. Wheat production is highly sensitive to climate, environmental changes and various abiotic stresses [4]. With increasing global food demand and the aggravation of abiotic stress, it is important to screen resistant and high-quality wheat varieties to maintain winter wheat yield and quality.

Salt stress is an important environmental factor affecting crop growth and yield. The accumulation of salt in soil can lead to osmotic pressure, which affects the absorption of water and the opening and closing of stomata, inhibiting normal growth [5,6]. Excessive salt accumulation in soil will also affect the yield and quality of winter wheat. The salt stress tolerance of plants largely depends on their roots. Expansin is a protein involved in cell wall relaxation, plant growth and development and response to plant diseases and other stresses. Han conducted a genome-wide analysis of wheat and identified 241 expansin genes, which were divided into three subfamilies: EXPA, EXPB and EXLA. Most wheat expansin genes were specifically expressed in the roots [7]. The expression of the *AtEXPA17* gene affected the formation of lateral roots in *Arabidopsis* [8]. The *AtEXLA2* gene is expressed in *Arabidopsis* lateral root crown cells, root elongation regions and different stages of lateral root development, and the overexpression of the *AtEXLA2* gene can promote the growth of the root hair of transgenic plants [9]. The overexpression of the *AtEXPA4* gene can promote the elongation of taproot in transgenic plants, while the deletion of the gene can reduce the growth rate of taproot [10,11].

As a regulatory factor in the plant cell wall, the expression of expansin is induced by the environment and plays a very important role in plant response to salt stress. Salt stress has adverse effects on plant metabolism through ionic toxicity, osmotic stress and oxidative stress [12]. When the salt concentration exceeds normal levels, the water potential in the soil will decrease, reducing the absorption of water by the plant roots [13,14]. A lack of water will cause water potential to decrease, swelling pressure to weaken, stomata to close and cell growth to slow down. Salt stress can also lead to the excessive production of reactive oxygen species (ROS), resulting in oxidative damage to plants [15]. Osmotic regulation is an important means for plants to maintain cell swelling pressure, which regulates plant growth and productivity by maintaining plant metabolic activity. Plants can promote osmotic balance at the cell level by synthesizing osmotic regulatory substances such as proline and soluble sugar, which reduce the production of reactive oxygen species to resist the adverse effects of salt stress. [16,17]. The overexpression of the *TaEXPA2* gene can enhance the tolerance of overexpressed plants to oxidative stress, salt stress and drought stress, which also promotes the formation of lateral roots of overexpressed plants, helping transgenic plants absorb water and nutrients [18,19,20]. The *OsEXPA7* gene can increase the tolerance of overexpressed rice to salt stress by coordinating the transport of Na^+^, scavenging ROS and relaxing the cell wall [21].

Our research group has also undertaken a lot of research on the expansin gene in winter wheat. We identified 128 coding sequences of extended proteins in the wheat genome, including twenty-six *TaEXPAs*, fifteen *TaEXPBs* and four *TaEXLAs*, and found that most *TaEXPs* were only expressed in the roots or multiple organs [22]. Furthermore, we overexpressed the winter wheat expansin genes *TaEXPA4-A/B/D*, *TaEXPA7-A/B/D*, *TaEXPA8-B/D*, *TaEXPA19* and *TaEXPB7-B* in *Arabidopsis* and found that these genes are closely related to the regulation of plant root growth, and they also have a certain role in resisting adversity stress [23,24,25]. In order to further screen the EXPA gene in winter wheat, which is closely related to the regulation of root growth, this study used Arabidopsis strains that overexpressed the *TaEXPAs* gene in winter wheat as materials. Based on the morphological observation and quantitative statistics of roots of different strains, combined with the bioinformatics analysis results of five plant EXPA family members, we finally screened the high-quality gene *TaEXPA7-B* related to root development in winter wheat. In addition, the function of this gene on root growth and the regulation of salt stress tolerance was further identified in rice, which provided the relevant theoretical basis and practical reference for further screening and application of high-quality expansin protein genes regulating root development in winter wheat.

## 2. Results

### 2.1. Morphological Observation of Arabidopsis Roots Overexpressing TaEXPAs

#### 2.1.1. Analysis of *Arabidopsis* Root Hairs Overexpressing *TaEXPAs*

In order to clearly observe the root growth state of plants overexpressing *TaEXPA4-A/B/D*, *TaEXPA7-A/B/D*, *TaEXPA8-B/D*, *TaEXPA19-A/D* and wild-type *Arabidopsis*, root tips of 3-day-old plants were taken to observe the growth state of root hairs in this study. The root hair growth state of *Arabidopsis* overexpressing the *TaEXPA7-B* gene was significantly better than that of wild-type and other transgenic plants. The average number of root hairs in wild-type Arabidopsis was 71, which ranged from 84 to 136 in overexpressed *Arabidopsis* (Figure 1).

#### 2.1.2. Analysis of *Arabidopsis* Root Length Overexpressing *TaEXPAs*

The seeds of the wild-type and overexpressed *TaEXPA4-A/B/D*, *TaEXPA7-A/B/D*, *TaEXPA8-B/D* and *TaEXPA19-A/D Arabidopsis* were sterilized and planted in vertical plates. The root growth of each genotype was observed after 7 days (Figure 2). The results showed that the root length of overexpressed *Arabidopsis* was significantly longer than that of the wild type. The average root length of the wild type was 1.3 cm, while the root length of *Arabidopsis* overexpressing *TaEXPA7-B* and *TaEXPA8-B* ranged from 1.59 cm to 2.63 cm. All of these results demonstrated that *TaEXPA7-B* and *TaEXPA8-B* genes could significantly promote the root length of *Arabidopsis*.

#### 2.1.3. Analysis of *Arabidopsis* Lateral Root Overexpressing *TaEXPAs*

In order to observe the lateral root growth of wild-type and overexpressed *TaEXPA4-A/B/D*, *TaEXPA7-A/B/D*, *TaEXPA8-B/D* and *TaEXPA19-A/D Arabidopsis*, Arabidopsis seeds of the above genotypes were sterilized and planted in vertical plates containing 1/2 MS solid medium. After 7 days, the tallest and shortest plants were removed, and the *Arabidopsis* of each genotype was photographed (Figure 3). The results showed that the lateral roots of transgenic *Arabidopsis* were thicker than those of the wild type. The average number of lateral roots in wild-type Arabidopsis was 36, and the overexpressed *Arabidopsis* ranged from 48 to 78. The number of lateral roots in *Arabidopsis* overexpressing the *TaEXPA7-B* gene was the highest, with 78.

#### 2.1.4. Analysis of *Arabidopsis* Root Cell Overexpressing *TaEXPAs*

We inoculated wild-type and overexpressed *TaEXPA4-A/B/D*, *TaEXPA7-A/B/D*, *TaEXPA8-B/D* and *TaEXPA19-A/D* genes of the *Arabidopsis* seeds in 1/2 MS medium and observed the root growth of each genotype plant after 7 days. We made temporary sections from the mature zone of *Arabidopsis* roots and observed the size of *Arabidopsis* root cells under an electron microscope (Figure 4). The average length of root cells in wild-type *Arabidopsis* is 52.71 μm. The root cell length of *Arabidopsis* overexpressing various genotypes ranges from 60.4 to 78.26 μm; the average width of root cells in wild-type *Arabidopsis* is 41.12 μm. The root cell width of *Arabidopsis* overexpressing various genotypes ranges from 45.44 to 52.85 μm. The length and width of root cells are significantly higher than those of the wild type. Moreover, the root cell length and width values of *Arabidopsis* overexpressing the *TaEXPA7-B* gene were the highest at 78.26 and 52.85 μm, respectively. The results indicate that the *TaEXPA7-B* gene has the most significant effect on root cell growth.

### 2.2. Bioinformatics Analysis

#### 2.2.1. Phylogenetic Analysis of the EXPA Family Genes in Five Plants

The phylogenetic tree shows the homologous relationship between genes. In order to confirm the evolutionary relationship among the EXPA family genes, the amino acid sequences of 192 EXPA family genes in these five plants were aligned and analyzed to establish rootless trees (Figure 5). Phylogenetic tree analysis showed that the EXPA family genes of the five plants were clustered separately; *Physcomitrella patens*, *Selaginella moellendorffii* and a few *Arabidopsis thaliana* clustered; *Picea abies*, *Triticum aestivum* and some *Arabidopsis thaliana* clustered together. The difference in the number of genes clustered in each group suggests that there may be evolutionary differences in the EXPA family genes among different plant species. There are 33 in *Physcomitrella patens*, 30 in *Selaginella moellendorffii*, 41 in *Picea abies*, 62 in *Triticum aestivum* and 26 in *Arabidopsis thaliana*.

#### 2.2.2. Promoter Analysis of EXPA Family Genes in Five Plants

In order to screen the genes related to root development in the EXPA family genes, we predicted cis-acting elements in promoters of EXPA family genes in five plants by PlantCARE and PLACE, including those related to plant growth and development, hormones and stresses (Figure 6). Most of the EXPA family genes also contained root growth and development-related elements (RHEs), and the *TaEXPA7-B* gene contained the most RHE elements (10). Therefore, the *TaEXPA7-B* gene was presumed to play an important role in plant root growth and development.

### 2.3. Subcellular Localization of TaEXPA7-B Protein

We observed the inner epidermal cells of onions infected with *Agrobacterium* using a fluorescence microscope (Figure 7). Under the infection of *Agrobacterium tumefaciens*, the whole inner epidermal cells of the onion emit green fluorescence, indicating that the eGFP protein is expressed in whole cells. Under the infection of *Agrobacterium* containing pCambia2300::TaEXPA7-B plasmid, green fluorescence can be observed on the cell wall of the inner epidermis of the onion, but there is no green fluorescence in the cytoplasm and nucleus. We conducted a plasma wall separation test on the inner epidermal cells of onion and found that the cell wall emits green fluorescence, but there is no green fluorescence on the cell membrane. The results indicate that the TaEXPA7-B protein is localized on the cell wall.

### 2.4. Identification of Rice Overexpressing TaEXPA7-B Gene

The transcription of the *TaEXPA7-B* gene in plants was detected using the RT-PCR method (Figure 8). The results indicated that the *TaEXPA7-B* gene has been successfully transcribed in transgenic rice. Based on the results, OE-2, OE-3 and OE-6 (rice lines 2, 3 and 6 overexpressing the *TaEXPA7-B* gene) were selected as the next research subjects.

The protein expression levels of rice overexpressing the *TaEXPA7-B* gene were identified. HA antibodies were used to detect whether the TaEXPA7-B protein was already expressed in transgenic rice, but the imprint color in OE-3 was lighter, which may be due to the lower expression level of the TaEXPA7-B protein in OE-3 lines.

### 2.5. Phenotypic Observation of Rice Overexpressing TaEXPA7-B Gene

#### 2.5.1. External Structure of Rice Overexpressing *TaEXPA7-B* Gene

We measured the plant height, root length and root number of 15-day-old overexpressed *TaEXPA7-B* gene and wild-type rice (Figure 9). The results showed that the plant height of OE-2/3/6 was significantly higher than that of wild-type rice. The average plant height of wild-type rice was 44.99 cm, while OE-2, OE-3 and OE-6 plants were 52.29 cm, 55.54 cm and 55.33 cm, respectively. Root length and number of roots of OE-2/3/6 were significantly higher than those of the wild type. The average root length of wild-type rice was 9.81 cm, and the lengths of OE-2, OE-3 and OE-6 plants were 13.79 cm, 13.36 cm and 14.05 cm, respectively. The average number of roots of wild-type rice was 17, and those of OE-2, OE-3 and OE-6 were 24 cm, 24 cm and 23 cm, respectively. All of the results showed that the *TaEXPA7-B* gene had a significant promotion effect on the root growth of overexpressing plants.

#### 2.5.2. Anatomical Structure of Rice Overexpressing *TaEXPA7-B* Gene

Paraffin section experiments (longitudinal and transverse sections) were performed on the root tips of wild-type and overexpressed *TaEXPA7-B* gene rice (Figure 10). The results showed that the height, length and width of cortical cells in the roots of OE-2/3/6 were significantly higher than those of wild-type rice, and the number of cortical layers was also higher than that of wild-type rice. The average height of root cortex cells in wild-type rice is 69.85 μm. The average root length of OE-2, OE-3 and OE-6 plants are 80.46, 73.19, and 79.91 μm, respectively. The average length of root cortex cells in wild-type rice is 21.95 μm. The average length of root cortex cells in OE-2, OE-3 and OE-6 are 31.71, 30.98 and 29.65 μm, respectively. The average width of root cortex cells in wild-type rice is 24.53 μm. The average width of root cortex cells in OE-2, OE-3 and OE-6 are 29.67, 31.29 and 29.95 μm, respectively. The average number of cell layers in the root cortex of wild-type rice is nine, while the average number of cell layers in the root cortex of OE-2, OE-3 and OE-6 are twelve, ten and eleven, respectively. The average lateral root primordial number at the root tip of wild-type rice is 1, while the average lateral root primordial numbers of OE-2, OE-3 and OE-6 are 3, 2 and 3, respectively.

### 2.6. Salt Stress Tolerance of Rice Overexpressing TaEXPA7-B Gene

#### 2.6.1. Phenotype of Rice Overexpressing *TaEXPA7-B* Gene under Salt Stress

OE-2/3/6 and wild-type rice were treated with 250 mM NaCl solution for 7 days, and their survival status was observed (Figure 11). The results showed that OE-2/3/6 and wild-type rice showed different degrees of wilting and yellowing after 7 days of NaCl treatment, and the wilting rates of OE-2 and OE-6 were 79.62% and 72.63%, respectively, which were significantly lower than those of the wild type.

#### 2.6.2. Osmoregulatory Substance Content of Rice Overexpressing *TaEXPA7-B* Gene under Salt Stress

Under NaCl treatment, the soluble sugar content of wild-type rice showed a trend of first increasing and then decreasing and reached its highest value at 6 h of treatment, with a soluble sugar content of 8.18 mg/g (Figure 12A). The soluble sugar content of OE-2/3/6 increased, and the soluble sugar content was higher than that of the wild type in each treatment period, reaching the highest at 24 h of salt treatment, at 14.72, 12.47 and 15.13 mg/g, respectively. The soluble protein content of wild-type and OE-2/3/6 showed a trend of first increasing and then decreasing, and the soluble protein content of OE-2/3/6 was significantly higher than that of wild-type in each treatment period. The content of soluble protein in the wild type reached its highest value of 50.35 mg/g at 3 h and then decreased. The soluble protein content of OE-2/3/6 reached its highest level at 12 h of treatment, at 73.84, 70.07 and 75.44 mg/g, respectively (Figure 12B). Under salt stress, the proline content of all plants increased with time, and the proline content in OE-2/3/6 was higher than that in wild-type plants. There was no significant difference in proline content between OE-2/3/6 and wild-type at 0 h and 1 h. At 24 h of NaCl treatment, the content of wild-type proline was 22.59 μg/g, and that of OE-2/3/6 were 39.06, 37.06 and 41.73 μg/g, respectively (Figure 12C).

#### 2.6.3. Morphogenesis of Related Substance Content of Rice Overexpressing *TaEXPA7-B* Gene under Salt Stress

After NaCl treatment, the lignin content of wild-type and overexpressed plants changed a little (Figure 12D). The lignin content of wild-type and overexpressed plants reached the highest at 1 h, with 23.37 mg/g for wild-type plants and 36.33/32.99 and 32.33 mg/g for OE-2/3/6, respectively. Under normal conditions, overexpressed plants accumulate more cellulose than wild-type rice (Figure 12E). After NaCl treatment, the cellulose content of wild-type and overexpressed plants decreased. After 24 h treatment, the cellulose content decreased to the lowest level. The cellulose content of the wild type was 105.12 mg/g, and the cellulose contents of OE-2/3/6 were 119.14, 130.17 and 141.87 mg/g, respectively. After NaCl treatment, the hemicellulose content of wild-type rice had little change, but the cellulose contents of OE-2/3/6 showed an increasing trend in general and reached the highest values at 24 h, which were 111.95, 114.31 and 104.34 mg/g, respectively. Moreover, the hemicellulose content of OE-2/3/6 was higher than that of wild-type at each period after NaCl treatment (Figure 12F).

#### 2.6.4. Antioxidant Enzyme Activity of Rice Overexpressing *TaEXPA7-B* Gene under Salt Stress

After NaCl treatment, the SOD activity of all plants showed an increasing trend over time (Figure 12G). At different time periods, the SOD activity of rice overexpressing the *TaEXPA7-B* gene was higher than that of the wild type. When treated with NaCl for 24 h, the SOD activity of rice reached its highest value. The SOD activity values of OE-2/3/6 were 861.61, 662.79 and 778.27 U/g, respectively, which was 1.27–1.65 times higher than that of the wild type (522.62 U/g). The POD activity of plants overexpressing the *TaEXPA7-B* gene was higher than that of the wild type (Figure 12H). The POD activity of OE-2/3/6 showed an overall trend of first increasing and then decreasing, reaching the highest values of 715.08, 654.14 and 626.41 U/g at 3 h, respectively. The POD activity of the wild type reached its highest value of 467.7 U/g at 6 h and then decreased. Under salt stress, the CAT activity of OE-2/3/6 and the wild type showed a trend of first increasing and then decreasing. The CAT activities of OE-3/6 and OE-2 reached their highest values at 1 h and 3 h, with activities of 93.99, 94.32 and 91.52 nmol/min/g, respectively, and gradually decreased thereafter. Moreover, at each period, the CAT activity of rice overexpressing the *TaEXPA7-B* gene was significantly higher than that of the wild type (Figure 12I).

## 3. Discussion

Expansin is involved in the initiation of plant root hairs and lateral root formation, and different expansin family members have different expression patterns during root growth and development [26]. Studies have shown that expansin can promote root growth by reviving inactivated root cells in plants [27,28]. EXPA is the largest subfamily in most plants and plays an important role in the development of plant roots. *OsEXPA1*, *OsEXPA2*, *OsEXPA3* and *OsEXPA4* are specifically expressed in root tips and participate in root tip morphogenesis; *OsEXPA8*, *OsEXPA17*, *OsEXPA22* and *OsEXPA30* are specifically expressed in root hairs and participate in root hair formation [29,30,31,32]. *AtEXPA4*, *AtEXP14* and *AtEXPA17* are involved in the separation of cortical cells near root primordia and initiation of lateral root primordia [8,11,33,34]. According to our previous research results on expansin in winter wheat, overexpressing the *TaEXPA4-A/B/D* gene can promote the growth of taproot and lateral root of transgenic *Arabidopsis*; overexpressing the *TaEXPA7-A/B/D* gene can also promote the development of root, stem, leaf and fruit of transgenic *Arabidopsis*. The plant height, root length, the number of root hairs and the number of leaves in *Arabidopsis* overexpressing *TaEXPA19-A/D* gene were better than those of the wild type [23,24]. In this study, we compared the root length, the number of lateral roots, the number of root hairs and the size of root cells in *Arabidopsis* overexpressing *TaEXPA4-A/B/D*, *TaEXPA7-A/B/D*, *TaEXPA8-B/D* and *TaEXPA19-A/D* genes in winter wheat.

Multiple sequence alignment analysis can reveal similar relationships between sequences and infer homologous relationships among gene family members in terms of structure and function. Based on the previous research results of our research group, we obtained 192 genes from the EXPA family. The EXPA family genes of *Physcomitrella patens*, *Selaginella moellendorffii*, *Picea abies*, *Triticum aestivum* and *Triticum aestivum* are 33, 30, 41, 62 and 26, respectively. We constructed a phylogenetic tree of 192 expansins from the above five higher plants. From the perspective of plant evolution, the number of expansin genes is gradually increasing, which may be due to gene duplication in plants during the evolutionary process [35]. In plants, both environmental and internal factors can affect the expression patterns of expansin, enabling them to participate in various developmental processes that may involve the regulation of corresponding cis-acting elements. We conducted cis-acting element analyses on the promoter sequences of 192 expansin genes and found that many of them were related to plant hormones, growth and development and stress. The heat map showed that most of the 192 EXPA family gene promoters contain MeJA- and ABA-related cis-acting elements, suggesting that these genes may play a role in responding to jasmonic acid and abscisic acid signals. The promoters of EXPA family genes in five plants all contain cis-acting elements (RHEs) that are related to root growth and development. Moreover, the promoter of *TaEXPA7-B* contains the highest number of RHE elements, with a total of 10. Therefore, we infer that the *TaEXPA7-B* gene plays an important role in the growth and development of wheat roots.

Expansin participates in plant growth and development by relaxing the cell wall. During the development of plant organs, expansins are usually expressed in a tissue-specific manner. TaEXPB7-B and SoEXPA1 are localized to the cell wall [25,36]; OsEXPA17 and HvEXPB7 are localized to the cell membrane [31,37]. The TaEXPA7-B protein was localized in the cell wall of onion by transient transfection, which indicated that it may affect the growth and development of wheat roots by directly participating in the relaxation and extension of the plant cell wall.

In previous experiments, the *TaEXPA7-B* gene has been transferred into Arabidopsis to explore its function [24]. However, *TaEXPA7-B* is a monocotyledonous gene in wheat, while *Arabidopsis* belongs to dicotyledonous plants. It may need to be more representative to identify the function of genes in monocotyledonous plants using dicotyledonous model plants as materials. Therefore, in this study, monocotyledonous model plant rice was selected for genetic transformation to further explore the function of *TaEXPA7-B*, which belongs to the same family as winter wheat. Expansin increases root diameter and cell volume, suggesting that upregulated expansin genes may have a role in promoting horizontal root cell elongation [38]. The phenotypic observation results of rice overexpressing the *TaEXPA7-B* gene showed that the plant height, root length and the number of roots of overexpressing plants were significantly higher than those of the wild type. The results of the paraffin sectioning of the rice root tips showed that the height, length and width of root cortex cells of OE-2/3/6 were significantly higher than those of the wild type, and the number of cortex layers was also more. The height, length and width of cortical cells in the roots of overexpressed plants were 1.05–1.15 times, 1.35–1.44 times and 1.06–1.29 times those in the wild type, respectively. Therefore, the *TaEXPA7-B* gene may promote the growth of plant roots by regulating the volume and number of cortical cells. In the permanent section of the rice root tips, we also observed that the number of lateral roots of rice overexpressing the *TaEXPA7-B* gene was higher than that of the wild type, which was consistent with the expected results.

High salinity can negatively affect the growth and development of the plant and, ultimately, yield [39]. Salinity stress adversely affects plant metabolism through ionic toxicity, osmotic stress and oxidative stress [12]. When salt concentrations exceed normal levels, water potential in the soil decreases, reducing water uptake by plant roots [13,14]. The results of this experiment showed that the soluble protein, soluble sugar and proline contents of rice overexpressing the *TaEXPA7-B* gene were significantly higher than those of the wild type under salt stress. Under salt stress, overexpressed rice maintains osmotic pressure by accumulating a large number of osmotic regulators, such as soluble sugar, soluble protein and proline, thus reducing cell damage. Some studies have shown that expansin is related to the accumulation of lignin in plants. Under the condition of expansin-mediated relaxation of plant cell walls, enzymes related to lignification affect the substrate, resulting in the accumulation and transfer of lignin. Moreover, during the relaxation of plant cell walls, expansin can transfer lignin by using enzymes related to lignin as subunits, thus increasing lignin content [40]. The results of this study showed that rice overexpressing the *TaEXPA7-B* gene accumulated more lignin, cellulose and hemicellulose in roots than wild-type rice. Expansin also influences the activity of antioxidant enzymes in plants [20]. Under salt stress, higher antioxidant enzyme activity was observed in overexpressed *OsEXPA7* lines compared to wild-type plants [21]. The *GhEXLB2* gene can increase SOD and POD activities and reduce the MDA content of transgenic plants [36]. Maintaining antioxidant enzyme activity at a high level is helpful for plants in terms of clearing ROS from their own cells and stabilizing plasma membranes [41]. In this study, the activities of antioxidant enzymes (SOD, POD and CAT) in the roots of rice overexpressing the *TaEXPA7-B* gene were significantly higher than those of wild-type plants under salt stress. All of these results indicated that the *TaEXPA7-B* gene could improve salt tolerance by regulating the content of osmotic regulators and morphogenesis-related substances and antioxidant enzyme activities in overexpressed rice roots. Combined with the results of promoter analysis of EXPA family genes, we think that rice may regulate various antioxidant enzymes and other substances through the ABA or MeJA pathway to resist salt stress. When plants are exposed to salt stress, roots are the first to sense and adapt to it. Salinity stress can change plant root architecture and cell structure and reduce root biomass. At the same time, the plant roots will make changes at the molecular level to maintain normal plant growth by regulating ions, osmotic substances and antioxidant enzymes in the body. The *TaEXPA7-B* gene may play an important role in this process.

## 4. Materials and Methods

### 4.1. Bioinformatics Analysis of EXPA Family Genes

According to the previous research results of our laboratory [23], 192 EXPA family genes were obtained. There were 33 EXPA family genes in *Physcomitrella patens*, 30 in *Selaginella moellendorffii*, 41 in *Picea abies*, 62 in *Triticum aestivum* and 26 in *Arabidopsis thaliana*. We used ClustalW to perform alignment analysis on five plant protein sequences and imported the results into MEGA7 to construct a phylogenetic tree. In this paper, the maximum likelihood tree was chosen to show the homologous relationships among EXPA family proteins in five plant species. Genome and protein databases of five plants were downloaded from websites: *Physcomitrella patens* (http://www.phytozome.net/physcomitrella (accessed on 20 October 2020), *Selaginella moellendorffii* (http://genome.jgi-psf.org/Selmo1/Selmo1.download.ftp.html (accessed on 20 October 2020), *Picea abies* (http://congenie.org/ (accessed on 21 October 2020), *Triticum aestivum* (http://ensemblgenomes.org (accessed on 21 October 2020) and *Arabidopsis thaliana* (www.arabidopsis.org (accessed on 22 October 2020). The 2.0 kb promoter sequences of EXPA family members were predicted by promoter online analysis and prediction software. (PlantCARE (http://bioinformatics.psb.ugent.be/webtools/plantcare/html/ (accessed on 23 October 2020) and PLACE (https://www.dna.affrc.go.jp/PLACE/?action=newplace (accessed on 24 October 2020)) Excel 2010 software was used to produce thermal maps.

### 4.2. Morphological Observation of Arabidopsis Root System

Colombia wild-type *Arabidopsis* and *Arabidopsis* (T2) overexpressing the *TaEXPAs* gene were provided by the Department of Plant Resources and Molecular Biology, School of Life Sciences, Northeast Agricultural University. Seeds of transgenic Arabidopsis with *TaEXPA4-A/B/D*, *TaEXPA7-A/B/D*, *TaEXPA8-B/D*, *TaEXPA19-A/D* genes and wild-type *Arabidopsis* were sterilized with 75% alcohol for 30 s and 10% sodium hypochlorite for 15 min, respectively. Culture conditions: light 25 °C/16 h, dark 22 °C/8 h; light intensity: 6000 lx; and relative humidity: 80%. After 3 days, all genotypes of Arabidopsis were placed on glass slides. Additionally, we dropped 100 μL sterile water on the roots of the plants to stretch the root hairs and then made temporary sections. We observed the growth status of root hairs in various genotypes of *Arabidopsis* under a microscope. After 7 days, we measured the root length, lateral root number and root cell size of *Arabidopsis*.

### 4.3. Subcellular Localization of TaEXPA7-B Protein in Onion

The onion inner epidermis was cut into 1 cm^2^ squares and cultured on MS solid medium for 24 h at 28 °C without light. The pCambodia 2300::TaEXPA7-B::eGFP expression vector was constructed using the double enzyme digestion method. Moreover, pCambodia 2300::TaEXPA7-B::eGFP and pCambodia 2300::eGFP were transformed into *Agrobacterium tumefaciens* LBA4404 by liquid nitrogen rapid thawing method, then transiently transfected into onion inner epidermal cells. After 2 days of dark cultivation at 28 °C, we observed the inner epidermis cells of the transformed onion with a fluorescence microscope. Restriction endonucleases BamHI and SalI were purchased from NEB (New England Biolabs, Beijing, China). The fluorescence microscope model is BK-FL(Chongqing Aote Optical Instrument Co., Ltd., Chongqing, China).

### 4.4. Overexpression of TaEXPA7-B Gene in Rice

The seeds of rice were purchased from Pujie Biology. We constructed pCambodia 2301::TaEXPA7-B expression vector using the double restriction enzyme digestion method and transformed it into LBA4404 through the freeze–thaw process. Additionally, we infected rice callus tissue with transformed Agrobacterium and added 50 mg/L Kan to the culture medium for the screening of positive plants. We identified the transformation status of the *TaEXPA7-B* gene through the use of the RT-PCR method. The specific operational steps for rice genetic transformation referred to Zhou Yan’s method [42]. A HA tag was added to the C-terminus of the *TaEXPA7-B* gene to detect TaEXPA7-B protein levels. The total protein was extracted from rice roots, and Western blot was used to identify the protein expression levels of the *TaEXPA7-B* gene in the overexpressed rice. Restriction endonucleases BamHI and KpnI were purchased from NEB (New England Biolabs, Beijing, China). The antibody of TaEXPA7-B-HA is the Anti-HA-Tag Mouse mAb. The internal reference protein is the Actin (2P2) Mouse. The second antibody is the Goat Anti-Mouse IgG HRP. All antibodies were purchased from Abmart Medical Technology (Shanghai, China) Co., Ltd. The target gene and reference gene identified by RT-PCR are 777 bp and 192 bp, respectively. The target protein and reference protein are 26 kDa and 42 kDa, respectively.

### 4.5. Phenotype Observation of Rice Overexpressing TaEXPA7-B Gene

We soaked rice seeds overexpressing the *TaEXPA7-B* gene (T1) and wild-type rice seeds in 75% alcohol for 1 min and then washed them with sterile water 2 times, 10% sodium hypochlorite for 20 min, and sterile water for 5 times, and then added MS solid culture medium for culture. Culture conditions: light 30 °C/16 h, dark 28 °C/8 h; light intensity: 2500 lx; and relative humidity: 70%. After 45 days of culture, the average plant height, root length and root number of the wild-type and overexpressed plants were counted. Fifteen seedlings were randomly counted from each line. The root tips of the wild-type and overexpressed rice were taken for a paraffin section experiment (transverse section, longitudinal section) to observe the microstructure of the root system. The paraffin sectioning method referred to Chen’s method [43].

### 4.6. Salt Treatment and Physiological Index Determination of Rice Overexpressing TaEXPA7-B Gene

We selected 30-day-old wild-type and overexpressed *TaEXPA7-B* rice root segments for subsequent experiments. Rice was treated with 250 mM NaCl solution for 0 h, 1 h, 3 h, 6 h, 12 h and 24 h, respectively. Then, we took 0.1g of the plant roots, ground them in liquid nitrogen, and stored them in a refrigerator at −80 °C. Wild-type and overexpressed *TaEXPA7-B* rice plants were treated with 250 mM NaCl solution for 7 days, and the wilting rate was calculated according to the number of drooping and yellowing of the leaves (N = 100). Osmosis-related substances included soluble sugar (KT-2-Y), soluble protein (BCAP-2-W) and proline (PRO-2-Y). Morphosis-related substances included lignin (MZS-2-G), cellulose (CLL-2-Y) and hemicellulose (BXW-2-G). The activities of antioxidant enzymes included SOD (SOD-1-Y), POD (POD-1-Y) and CAT (CAT-1-Y). The above indicators were measured using the kit of Suzhou Comin Biotechnology Company (Suzhou, China), all shown as the content in fresh weight.

### 4.7. Statistical Analysis

All of the experiments were repeated at least 3 times. Data were presented as mean ± SD, plotted using GraphPad Prism 5 software and analyzed for significance of differences.

## 5. Conclusions

In the present study, the expansin gene *TaEXPA7-B*, which is closely related to the growth and development of plant roots, was screened from the EXPA family of five plants. The TaEXPA7-B protein is located in the cell wall, which may regulate root morphogenesis by acting on the plant cell wall. The expression of the *TaEXPA7-B* gene responds to salt stress, and the overexpression of this gene in rice significantly promotes the growth of plant roots and their tolerance to salt stress.

## Figures and Tables

**Figure 1 ijms-25-07707-f001:**
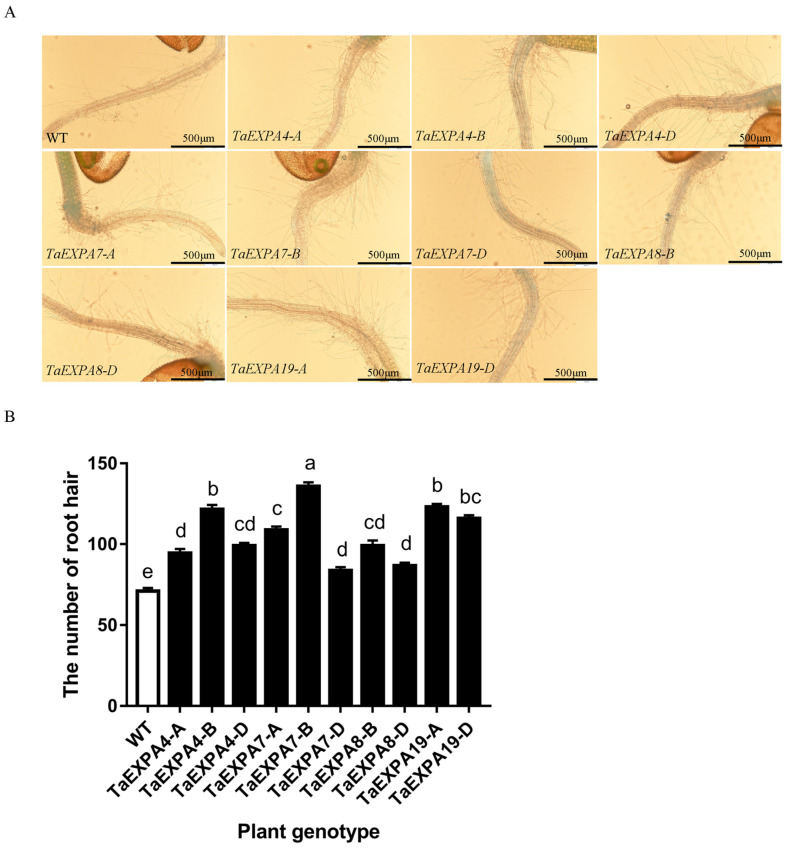
Root hairs of overexpressed and wild-type *Arabidopsis*. (**A**) Observation on root hairs of overexpressed *TaEXPA4-A/B/D*, *TaEXPA7-A/B/D*, *TaEXPA8-B/D* and *TaEXPA19-A/D* genes and wild-type *Arabidopsis*; bar = 500 um. (**B**) Statistics on the number of root hairs of overexpressed *TaEXPA4-A/B/D*, *TaEXPA7-A/B/D*, *TaEXPA8-B/D* and *TaEXPA19-A/D* genes and wild-type *Arabidopsis*. The letters (a–e) represent significant differences according to Tukey’s test (*p* < 0.05).

**Figure 2 ijms-25-07707-f002:**
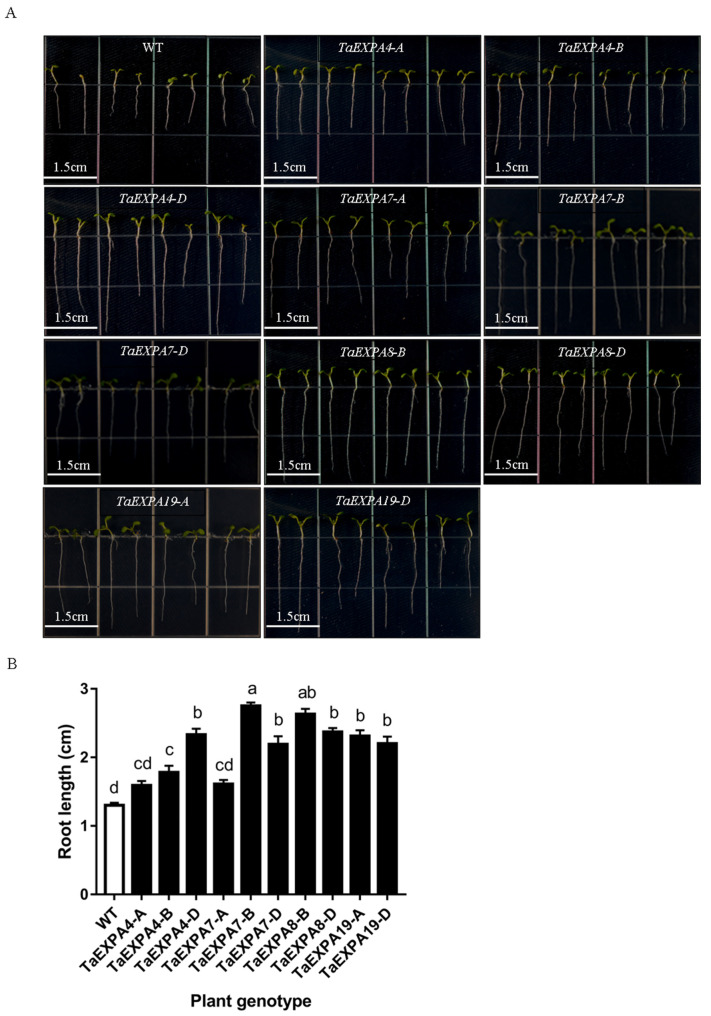
Root length of overexpressed and wild-type *Arabidopsis*. (**A**) Observation on root length of overexpressed *TaEXPA4-A/B/D*, *TaEXPA7-A/B/D*, *TaEXPA8-B/D* and *TaEXPA19-A/D* genes and wild-type *Arabidopsis*; bar = 1.5 cm. (**B**) Statistics on root length of overexpressed *TaEXPA4-A/B/D*, *TaEXPA7-A/B/D*, *TaEXPA8-B/D* and *TaEXPA19-A/D* genes and wild-type *Arabidopsis*. The letters (a–d represent significant differences according to Tukey’s test (*p* < 0.05).

**Figure 3 ijms-25-07707-f003:**
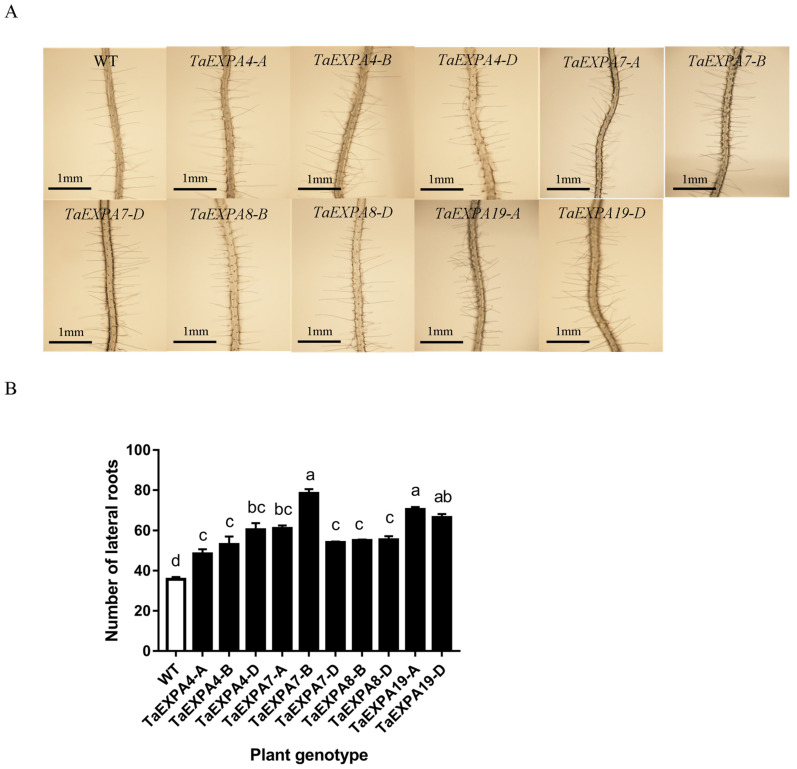
Lateral root of overexpressed and wild-type *Arabidopsis*. (**A**) Observation on lateral root of overexpressed *TaEXPA4-A/B/D*, *TaEXPA7-A/B/D*, *TaEXPA8-B/D* and *TaEXPA19-A/D* genes and wild-type *Arabidopsis*; bar = 1 mm. (**B**) Statistics on the number of lateral roots of overexpressed *TaEXPA4-A/B/D*, *TaEXPA7-A/B/D*, *TaEXPA8-B/D* and *TaEXPA19-A/D* genes and wild-type *Arabidopsis*. The letters (a–d) represent significant differences according to Tukey’s test (*p* < 0.05).

**Figure 4 ijms-25-07707-f004:**
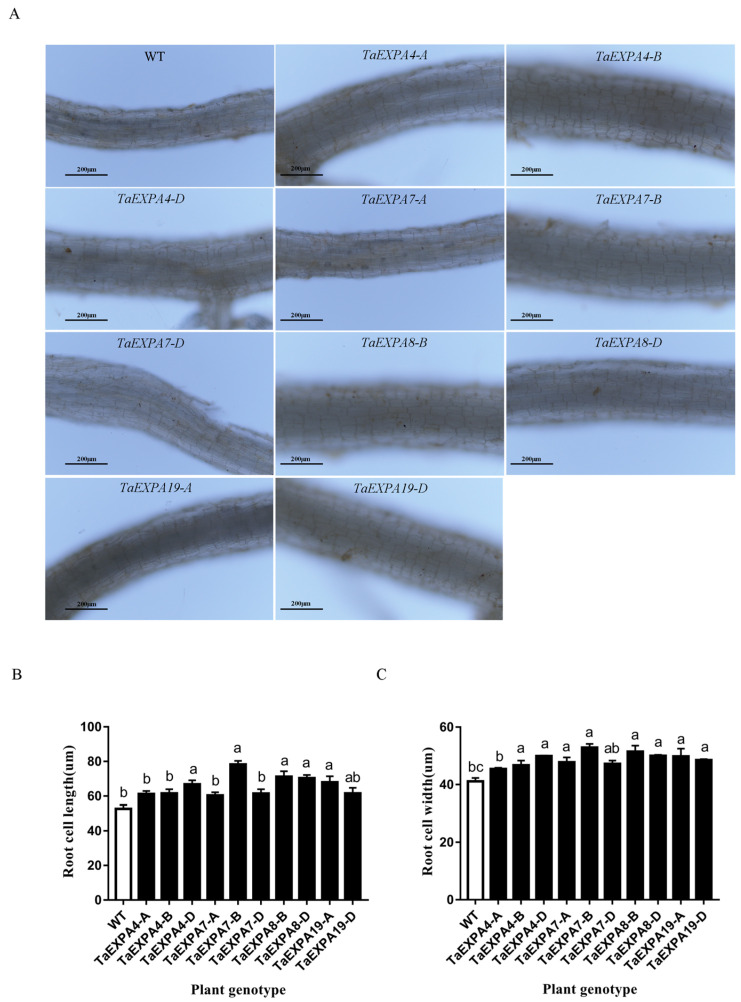
Root cell size of overexpressed and wild-type *Arabidopsis*. (**A**) Observation on root cell size of overexpressed *TaEXPA4-A/B/D*, *TaEXPA7-A/B/D*, *TaEXPA8-B/D* and *TaEXPA19-A/D* genes and wild-type *Arabidopsis*; bar = 200 um. (**B**) Statistics on the number of root cell length of overexpressed *TaEXPA4-A/B/D*, *TaEXPA7-A/B/D*, *TaEXPA8-B/D* and *TaEXPA19-A/D* genes and wild-type *Arabidopsis*. (**C**) Statistics on the number of root cell width of overexpressed *TaEXPA4-A/B/D*, *TaEXPA7-A/B/D*, *TaEXPA8-B/D* and *TaEXPA19-A/D* genes and wild-type *Arabidopsis*. The letters (a–c) represent significant differences according to Tukey’s test (*p* < 0.05).

**Figure 5 ijms-25-07707-f005:**
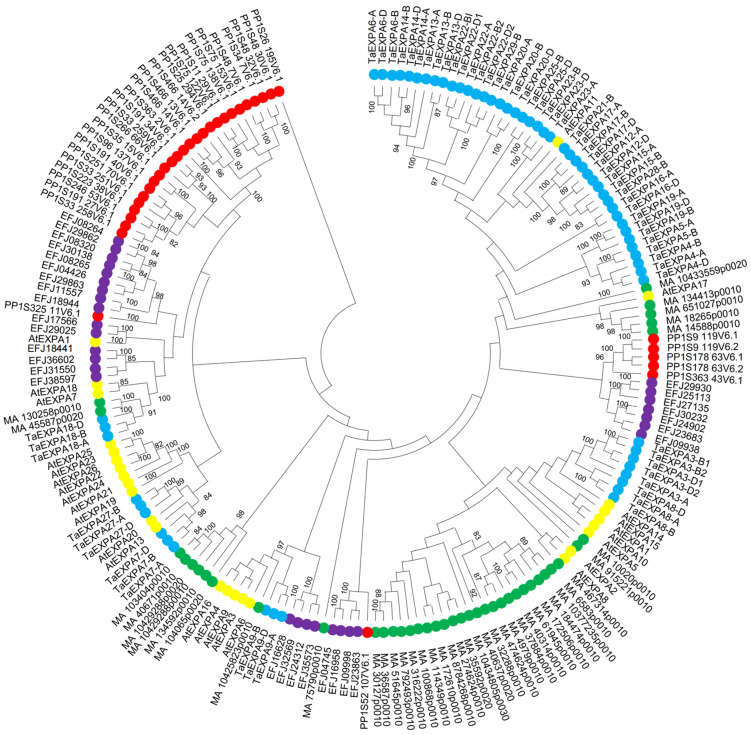
Amino acid phylogenetic tree of EXPA family members in five plants constructed using the maximum likelihood algorithm via MEGA7.0 software with 1000 Bootstrap replicates. The red represents the *Physcomitrella patens*; purple represents *Selaginella moellendorffii*; green represents *Picea abies*; blue represents *Triticum aestivum*; yellow represents *Arabidopsis thaliana*.

**Figure 6 ijms-25-07707-f006:**
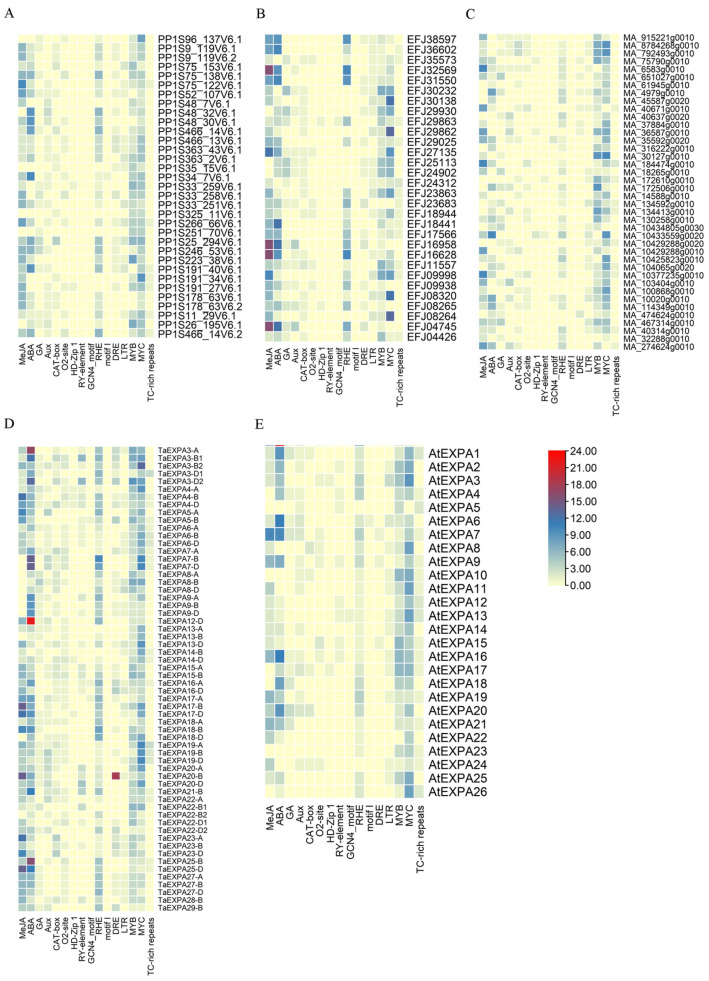
Promoter elements of the EXPA family genes in five plants. The number of promoters of EXPA family genes in five plants was counted to produce the heat map: (**A**) *Physcomitrella patens*, (**B**) *Selaginella moellendorffii*, (**C**) *Picea abies*, (**D**) *Triticum aestivum* and (**E**) *Arabidopsis thaliana*.

**Figure 7 ijms-25-07707-f007:**
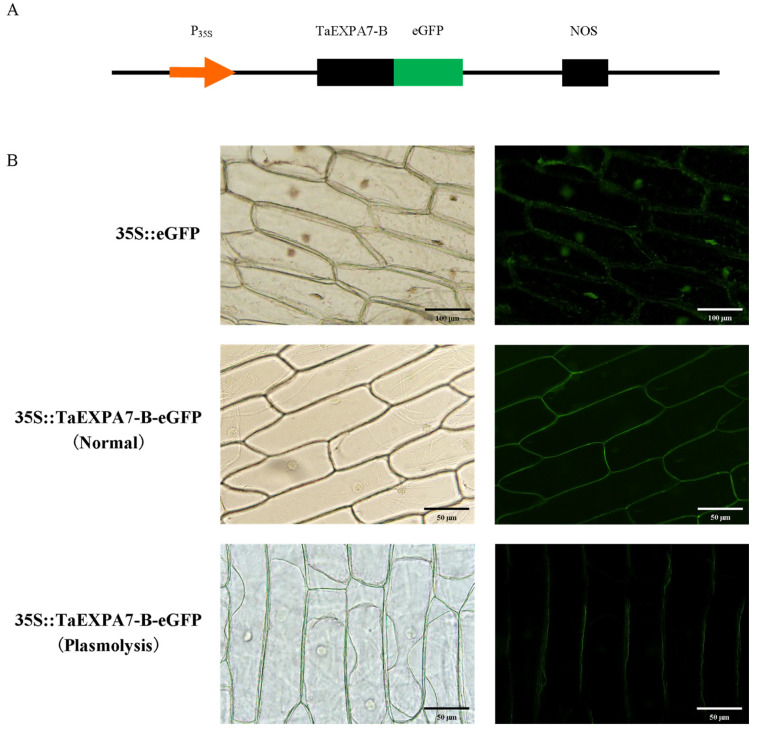
Subcellular localization results of TaEXPA7-B protein. (**A**) TaEXPA7-B expression vector construction for onion inner epidermal cells subcellular localization test. P_35S_ indicates CaMV35S promoter; eGFP indicates enhanced green fluorescent protein; and NOS indicates terminator. (**B**) TaEXPA7-B subcellular localization results; bar = 50 um.

**Figure 8 ijms-25-07707-f008:**
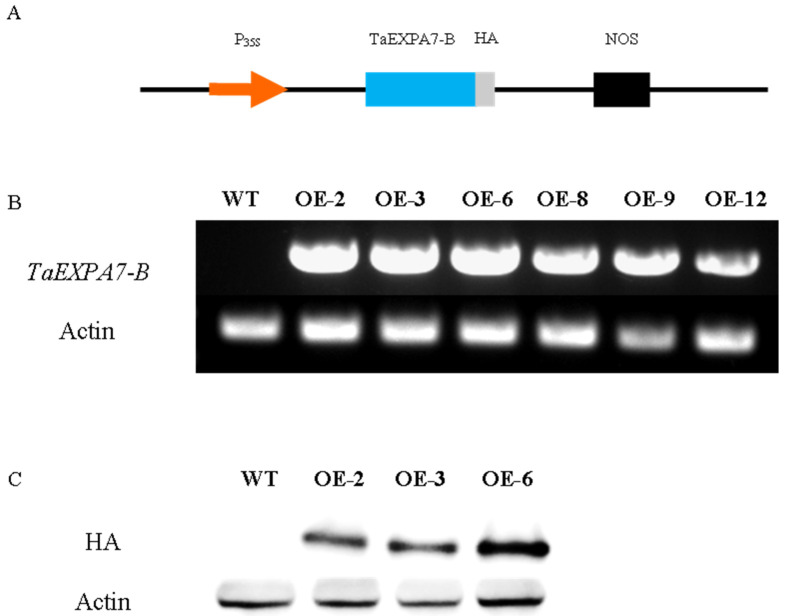
Identification of overexpressed rice. (**A**) Construction of TaEXPA7-B expression vector. P35S indicates CaMV35S promoter; NOS indicates terminator. (**B**) Identification of *TaEXPA7-B* in wild-type and overexpressed rice. (**C**) Identification of TaEXPA7-B protein in wild-type and overexpressed rice.

**Figure 9 ijms-25-07707-f009:**
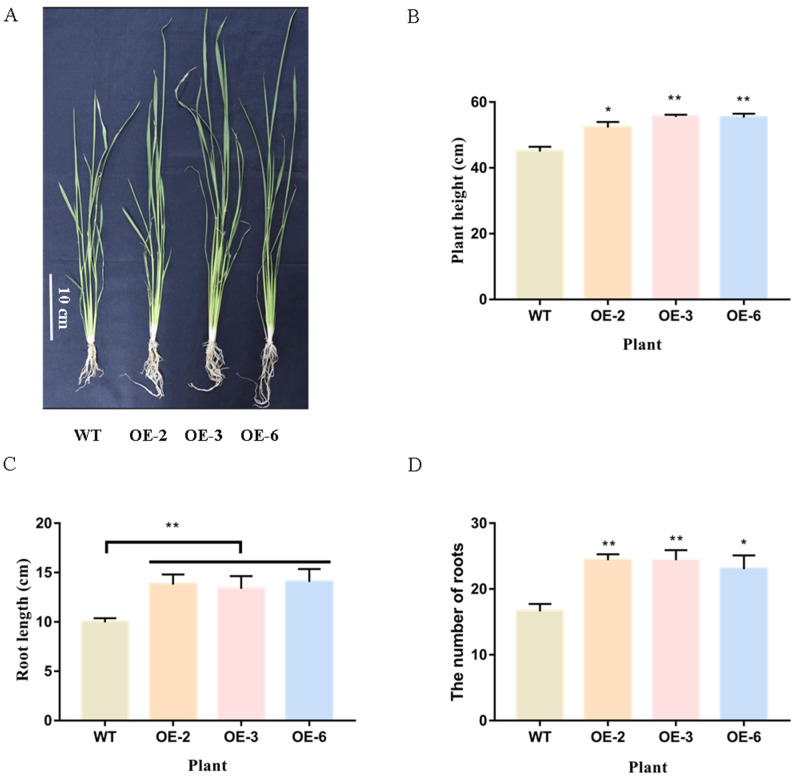
External structure of rice overexpressing *TaEXPA7-B* gene. (**A**) Phenotypes of wild-type and overexpressed *TaEXPA7-B* gene rice at 15 days. (**B**) Measurements of the plant height of wild-type and overexpressed rice. (**C**) The plant length of wild-type and overexpressed rice. (**D**) The number of roots of wild-type and overexpressed rice. * indicates significant differences in *t*-tests (* *p* < 0.05, ** *p* < 0.01).

**Figure 10 ijms-25-07707-f010:**
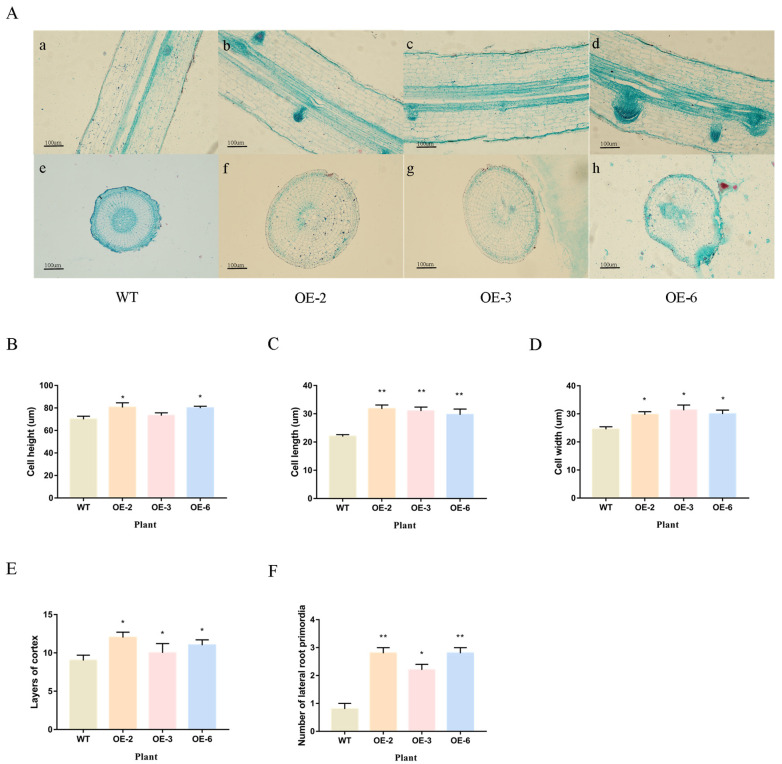
Observation and quantitative statistics of anatomical structure of wild-type and overexpressed *TaEXPA7-B* gene rice. (**A**) The observation of microstructure between wild-type and overexpressed *TaEXPA7-B* gene rice; a, b, c and d are the longitudinal cutting of rice roots; e, f, g and h are the cross-sections of rice roots. (**B**) Measurements of the cell height of rice root cortex. (**C**) The cell length of rice root cortex. (**D**) The cell width of rice root cortex. (**E**) The layers of rice root cortex. (**F**) The number of lateral root primordia. * indicates significant differences in *t*-tests (* *p* < 0.05, ** *p* < 0.01).

**Figure 11 ijms-25-07707-f011:**
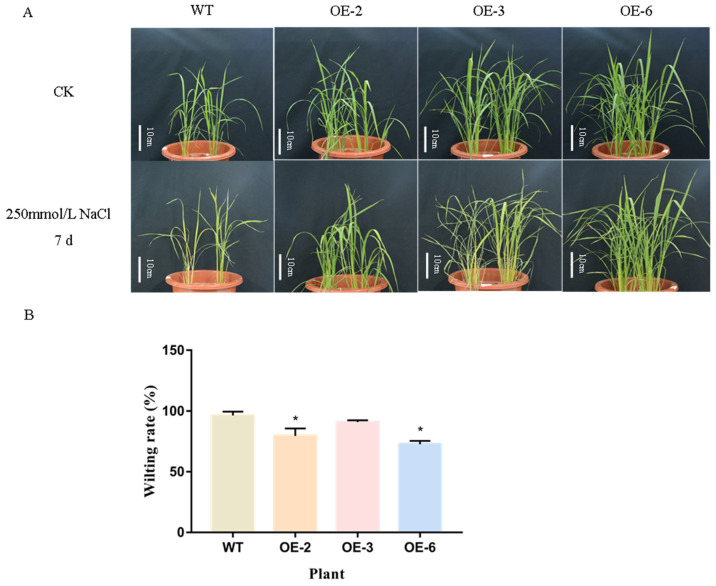
Tolerance of overexpressed *TaEXPA7-B* rice to salt stress. (**A**) Phenotype of rice overexpressing *TaEXPA7-B* gene under salt stress. (**B**) The wilting rate of rice leaves under salt stress. * indicates significant differences in *t*-tests (* *p* < 0.05).

**Figure 12 ijms-25-07707-f012:**
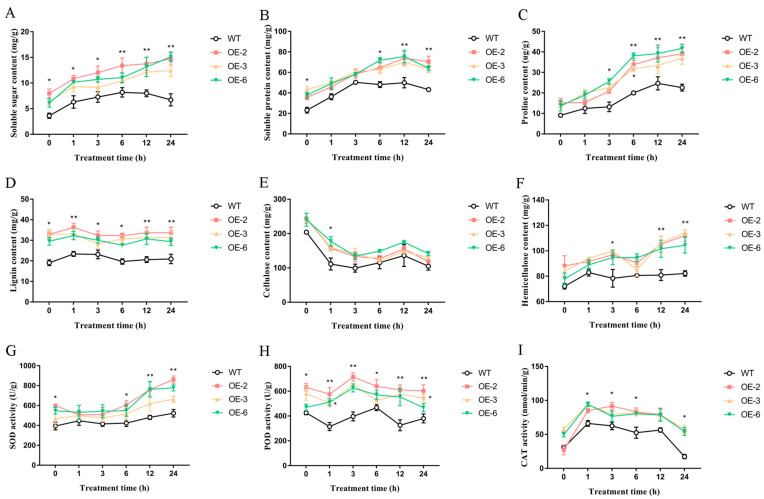
Changes in related substances content in overexpressed rice under salt stress. (**A**) Soluble sugar content. (**B**) Soluble protein content. (**C**) Proline content. (**D**) Lignin content. (**E**) Cellulose content. (**F**) Hemicellulose content. (**G**) Activity of SOD. (**H**) Activity of POD. (**I**) Activity of CAT. * indicates significant differences in *t*-tests (* *p* < 0.05, ** *p* < 0.01).

## Data Availability

All data are included in this paper.

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
