# Peer review of "A Study on the Functional Identification of Overexpressing Winter Wheat Expansin Gene TaEXPA7-B in Rice under Salt Stress"

_ijms, 2024, doi:10.3390/ijms25147707_

Round 1

Reviewer 1 Report

Comments and Suggestions for Authors

The study was focused on the functional Identification of overexpressing winter wheat expansin gene TaEXPA7-B in rice under salt stress.

The Authors quantified the morphology of the roots of Arabidopsis overexpressing the TaEXPAs gene in the early stage of development. Combined the bioinformatics analysis relating to EXPA genes in five plants and identified TaEXPA7-B, a member of the EXPA family closely related to root development in winter wheat. Subcellular localization analysis of the TaEXPA7-B protein showed that it is located in the plant cell wall. In the  study, the TaEXPA7-B gene was overexpressed in rice. The results showed that plant height, root length and the number of lateral roots of rice overexpressing the TaEXPA7-B gene were significantly higher than those of the wild type, and the expression of the TaEXPA7-B gene significantly promoted the growth of lateral root primordium and cortical cells. In addition, under salt stress, the accumulation of osmotic regulators, cell wall-related substances and the antioxidant enzyme activities of the overexpressed plants were higher than those of the wild type, and they had better salt tolerance.

The paper is quite interesting and important in the scientific field. However, some improvements are recommended:

-            The Introduction is very long and overloaded in the content. It should be rewritten into a more concise form.

-            Conversely, the Discussion is plain, and at least few aspects regarding the results may be considerably extended into a more in-depth interpretation.

-            Figure 2 – some photos of the roots are low graphical quality, so there is a problem to assess and compare the length of the roots. Therefore, increasing of the resolution is highly recommended.

-            Figure 12 contains too much detailed data, and the resolution is quite low.

-            Moderate editing of English language is required.

Comments on the Quality of English Language

Moderate editing of English language is required.

Author Response

  1. Response to comment:The Introduction is very long and overloaded in the content. It should be rewritten into a more concise form.

Response: The introduction has been shortened according to the reviewers' comments (Line 37-94).

2. Response to comment:Conversely, the Discussion is plain, and at least few aspects regarding the results may be considerably extended into a more in-depth interpretation.

Response: The discussion section has been revised according to the reviewers' comments.

3. Response to comment:Figure 2 – some photos of the roots are low graphical quality, so there is a problem to assess and compare the length of the roots. Therefore, increasing of the resolution is highly recommended.

Response: According to the reviewer's suggestion, we improved the resolution of Figure 2 (Line 209).

4. Response to comment:Figure 12 contains too much detailed data, and the resolution is quite low.

Response: We have reorganized Figure 12 according to the reviewer's comment and improved its resolution (Line 386).

5. Response to comment:Moderate editing of English language is required.

Response: Thank you for your valuable and thoutful comments. We tried our best to improve the manuscript and made some changes to the manuscript. These changes will not influence the content and framework of the paper. And here we did not list the changes but marked in red in the revised paper.

Reviewer 2 Report

Comments and Suggestions for Authors

Authors studied the expansin gene family; phylogenetics, promoter cis-active elements distribution. Authors analysed the influence of overexpression of these genes on roots morphology and anatomy.

Moreover the one selected gene TaEXPA7-B was overexpressed in rice ant its subcellular localization in cell walls was confirmed by fluorescence microscope. The anatomy of roots was studied, moreover the salt stress resistance was tested in terms od concentration of soluble sugar, soluble proteins and activity of SOD and POD enzymes. Results do not fully support conclusions, also materials and methods are poorly described. Figures are of good quality.

Although study seems to be logically performed, there are some problems that should be resolved:

1.      In section 2.1 Authors write of overexpression of several TaEXPA genes. The overexpression was not conformed experimentally for example by qRT-PCR in presented work and earlier studies. Therefore it is hard to link observed changes to gene overexpression.  

2.      In section 2.4 Authors present qualitative RT-PCR results it is hard to drawn conclusions based on these general studies, it could be better to perform quantitative qRT-PCR to precisely characterize the overexpression level.

3.      Discussion- present more precisely molecular mechanisms or signaling pathways linking TaEXPA genes expression/overexpression and, root morphology/anatomy changes, adaptation to salt stress (SOD/POD activity, sugar, protein concentratopn etc).

4.      In section 4.3 provide names of restriction enzymes used for cloning, name of fluorescence microscope model, manufacturer name, country of origin

5.      In section 4,.4 there is no names of restriction enzymes, no quantitative RT-PCR, these methods should be presented. qRT-PCR experiments should be precisely described, as a guide use following article :

MIQE précis: Practical implementation of minimum standard guidelines for fluorescence-based quantitative real-time PCR experiments - PMC (nih.gov)

6.      Authors wrte of salt stress, however the rapid placement of plant material into the 0.25 M NaCl solution is rather a salt shock, change the text accordingly, see following reference:

Salt stress or salt shock: which genes are we studying? - PubMed (nih.gov)

7.      Authors previously published articles studying the root morphology in Arabidopsis, it is question on novelty of presented results

Other comments

Line 62-plant names write in italics

Line 111- add space

Line 92-94: Sentence has no sense, rewrite it :

In order to further screen the EXPA gene in winter wheat, which is closely related to the regulation of root growth, this study used Arabidopsis strains that overexpressed the TaEXPAs gene in winter wheat as materials.

Comments on the Quality of English Language

Minor editing of English language required.

Author Response

  1. Response to comment:In section 2.1 Authors write of overexpression of several TaEXPA genes. The overexpression was not conformed experimentally for example by qRT-PCR in presented work and earlier studies. Therefore it is hard to link observed changes to gene overexpression.

Response: All the over-expressed Arabidopsis involved in section 2.1 are the results obtained by our research group in the early stage, and have been successfully transformed. Moreover, related articles have been published in Chinese or English journals (Part of the data is unpublished). Therefore, we did not identify it by qRT--PCR.

(Peng, L.N.; Xu, Y.Q.; Wang, X.; et al. Overexpression of paralogues of the wheat expansin gene TaEXPA8 improves low-temperature tolerance in Arabidopsis. Plant Biology 2019, 21, 1119-1131.

Zhao, Z.Y.; Xu, Y.Q.; Dong, J.M.; Wu, J.W.; Peng, L.N.; Feng, X.; Feng, S.S.; Hu, B.Z.; Li, F.L. Genetic transformation and functional analysis of TaEXPA7 homologous gene of winter wheat in cold region (In Chinese). Journal of Triticeae Crops 2020, 40, 401-407.)

2. Response to comment:In section 2.4 Authors present qualitative RT-PCR results it is hard to drawn conclusions based on these general studies, it could be better to perform quantitative qRT-PCR to precisely characterize the overexpression level.

Response: qRT-PCR method can only express the original genes in plants relatively quantitatively. The TaEXPA7-B is a gene in wheat and does not exist in rice. The results of relative quantitative detection of this gene in rice are also inaccurate. Therefore, we used RT-PCR to identify rice. In addition, we also extracted the total protein of plants for identification, which can confirm the successful expression of TaEXPA7-B in rice.

3. Response to comment:Discussion-present more precisely molecular mechanisms or signaling pathways linking TaEXPA genes expression/overexpression and, root morphology/anatomy changes, adaptation to salt stress (SOD/POD activity, sugar, protein concentratopn etc).

Response: The discussion section has been revised according to the reviewers' comments.

4. Response to comment:In section 4.3 provide names of restriction enzymes used for cloning, name of fluorescence microscope model, manufacturer name, country of origin

Response: I think you may be talking about the content in section 2.3. These contents about names of restriction enzymes used for cloning, name of fluorescence microscope model, manufacturer name, and country of origin have been supplemented (Line 134-137).

5. Response to comment:In section 4,.4 there is no names of restriction enzymes, no quantitative RT-PCR, these methods should be presented. qRT-PCR experiments should be precisely described, as a guide use following article : MIQE précis: Practical implementation of minimum standard guidelines for fluorescence-based quantitative real-time PCR experiments - PMC (nih.gov)

Response: I think you may be talking about the content in section 2.4. The information about restriction endonucleases has been supplemented. Because there is no TaEXPA7-B gene in rice, we think this experiment is not suitable for identification by qRT-PCR (Line 149-153).

6. Response to comment:Authors wrte of salt stress, however the rapid placement of plant material into the 0.25 M NaCl solution is rather a salt shock, change the text accordingly, see following reference: Salt stress or salt shock: which genes are we studying? - PubMed (nih.gov)

Response: Thank you very much for the reviewer's suggestion. I have also read the literature you recommended carefully. But we think that high concentration salt treatment can also be called salt stress. When we designed the experiment to choose the salt concentration, we referred to some literatures. In these literatures, continuous treatment with high concentration of salt is also called salt stress. Such as, “EgSPEECHLESS Responses to Salt Stress by Regulating Stomatal Development in Oil Palm”. They used three levels of salts (100, 250, and 500 mM NaCl) to treat young palms for 14 days. Therefore, we think that this treatment can also be called salt stress.

7. Response to comment:Authors previously published articles studying the root morphology in Arabidopsis, it is question on novelty of presented results

Response: In the early stage, we all transformed the expansin gene from wheat into Arabidopsis thaliana, and now we transform the TaEXPA7-B gene into rice with a closer relationship to further identify its function. We think the manuscript is still innovative.

Reviewer 3 Report

Comments and Suggestions for Authors

The paper combines a huge amount of work, interesting results, with some major weaknesses and a very poor presentation. Therefore authors must perform a great effort on the manuscript and clarify many obscure points.

The weakest part is the one dealing with Arabidopsis (figures 1-4). Apparently authors are comparing 10 overexpression lines, with each one overexpressing a different expasin gene, but no information is given on how these lines were constructed, which promoter and terminator are driving the expression of the transgene, which is the marker gene or if the control has been transformed with the empty plasmid. Another major problem is that authors are just comparing a single line for each expansin. This is quite dangerous, as different transgenic lines could behave different. The usual strategy is to compare at least three transgenic lines, as authors do in the rice experiments. Please complete all the information related to the Arabidopsis lines and all the experiments performed. Include supplemental data or references explaining how the lines were selected and confirm that the phenotype is similar in different lines.  Figure 2 is also very poor, as diferences are hardly seen. Can you try to enhance the contrast or the quality?

Figure 5 and 6 are also quite problematic. I do not see the point of including figure 5 after you have presented all the work with Arabidopsis. It should fit better as figure 1, kind of introduction for the expansin family.

Figure 6: All the results presented in the current work are based in transgenic overexpression. This sentence does not make sense: " The results showed that most of the EXPA family genes contained MeJA and ABA-related elements, which may respond to MeJA and ABA signals. Most of the EXPA family genes also contained root growth and development-related elements (RHEs), and the TaEXPA7-B gene contained the most RHE elements (10). Therefore, the TaEXPA7-B gene was presumed to play an important role in plant root growth and development. "

A bioinformatic promoter analisys is just a preliminary result, and should be supported by expression data (qRT-PCR) which is not presented. I addition I do not see much difference with other expansins analyzed in the same figure. The selection of TaEXPAT-B gene could be justified by the Arabidopsis results, as in most experiments (figures 1-4) is giving the strongest phenotype. I suggest to delete this figure, because questions the strategy followed by the authors or the project design.

The rice experiments seem more coherent... but still I have found some problems. In figure 8A there is no indication of an HA tag in the vector, but in figure 8C there is a western against this tag. Please clarify. Was the tag included in the vector? Are different lines?

Also the materials and methods section of these experiments cast a lot of doubts.

 "A HA tag was added to the C-terminus of the TaEXPA7-B gene to detect TaEXPA7-B protein levels. After mixing the roots, stems, and leaves of rice, the total protein was extracted, and Western blot was used to identify the protein expression levels of the TaEXPA7-B gene in the 488 overexpressed rice. The internal reference protein is Actin (2P2) Mouse. "

Why authors mix different tissues and do not evaluate expression in each tissue? If the protein expressed by the transgene is not stable in roots this would compromise the interpretation of the results. Also: which antibody have used to detect the HA tag? Please include the information and the manufacturer. I also do not understand how the internal reference is a mouse protein. I assume that is a mistake and authors are referring to the antibody they are using. Please include in figure 8 a western blot from root protein and clarify all this messy information in materials and methods.

The paper also needs a thorough revision of english. The written is very poor and difficult to understand.

Minor points

Line 62: "rosa rugosa" in italics. 

Comments on the Quality of English Language

English needs to be completely revised. 

Author Response

  1. Response to comment:The weakest part is the one dealing with Arabidopsis (figures 1-4). Apparently authors are comparing 10 overexpression lines, with each one overexpressing a different expasin gene, but no information is given on how these lines were constructed, which promoter and terminator are driving the expression of the transgene, which is the marker gene or if the control has been transformed with the empty plasmid. Another major problem is that authors are just comparing a single line for each expansin. This is quite dangerous, as different transgenic lines could behave different. The usual strategy is to compare at least three transgenic lines, as authors do in the rice experiments. Please complete all the information related to the Arabidopsis lines and all the experiments performed. Include supplemental data or references explaining how the lines were selected and confirm that the phenotype is similar in different lines.  Figure 2 is also very poor, as diferences are hardly seen. Can you try to enhance the contrast or the quality?

Response: The 10 over-expressed lines in the manuscript are all the results of our previous research, the construction and transformation process of vectors have been published in Chinese and English journals respectively (Part of the data is unpublished). According to the previous research of our group, we found that the expression of EXPA family genes in wheat has a significant effect on the root growth of plants, so we observed and compared the root morphological characteristics of 10 strains of Arabidopsis obtained by our group in the previous period. In addition, according to the reviewer's comments, Figure 2 has been improved in resolution and reorganized.

(Peng, L.N.; Xu, Y.Q.; Wang, X.; et al. Overexpression of paralogues of the wheat expansin gene TaEXPA8 improves low-temperature tolerance in Arabidopsis. Plant Biology 2019, 21, 1119-1131.

Zhao, Z.Y.; Xu, Y.Q.; Dong, J.M.; Wu, J.W.; Peng, L.N.; Feng, X.; Feng, S.S.; Hu, B.Z.; Li, F.L. Genetic transformation and functional analysis of TaEXPA7 homologous gene of winter wheat in cold region (In Chinese). Journal of Triticeae Crops 2020, 40, 401-407.)

2. Response to comment:Figure 5 and 6 are also quite problematic. I do not see the point of including figure 5 after you have presented all the work with Arabidopsis. It should fit better as figure 1, kind of introduction for the expansin family.

Response: We think that Figures 5 and 6 are necessary. Because in Figure 1-4, we only observed the root morphological characteristics of Arabidopsis that overexpressed the EXPA family genes in wheat. We selected five plants from the evolutionary point of view, and screened cis-acting elements (RHE) related to root development by analyzing the promoters of EXPA family genes. Through the above experimental results, the TaEXPA7-B gene closely related to root development in EXPA family were screened out.

3. Response to comment:Figure 6: All the results presented in the current work are based in transgenic overexpression. This sentence does not make sense: " The results showed that most of the EXPA family genes contained MeJA and ABA-related elements, which may respond to MeJA and ABA signals. Most of the EXPA family genes also contained root growth and development-related elements (RHEs), and the TaEXPA7-Bgene contained the most RHE elements (10). Therefore, the TaEXPA7-B gene was presumed to play an important role in plant root growth and development. "

Response: RHE elements in promoters are related to the growth and development of roots, so we take the number of rhee elements in promoters of EXPA family genes as a condition for pre-screening. Based on the analysis of cis-acting elements in the promoter of EXPA family gene and the observation results of root morphology of Arabidopsis overexpressed in the early stage, TaEXPA7-B gene was finally selected and transformed into a model plant of Gramineae with a closer relationship for subsequent verification. Additionally, this sentence “The results showed that most of the EXPA family genes contained MeJA and ABA-related elements, which may respond to MeJA and ABA signals.” has been deleted according to the reviewer's suggestion. (Line 275-277).

4. Response to comment:A bioinformatic promoter analisys is just a preliminary result, and should be supported by expression data (qRT-PCR) which is not presented. I addition I do not see much difference with other expansins analyzed in the same figure. The selection of TaEXPAT-B gene could be justified by the Arabidopsis results, as in most experiments (figures 1-4) is giving the strongest phenotype. I suggest to delete this figure, because questions the strategy followed by the authors or the project design.

Response: In our previous experiments, we found that overexpression of EXPA family genes in wheat can significantly promote the root growth of transgenic Arabidopsis. Therefore, we first compared and analyzed the roots of different genotypes of overexpressed Arabidopsis in our research group, and found that the TaEXPA7-B gene has a strong effect on promoting the root growth of transgenic plant. We want to further identify the gene closely related to root growth and development in EXPA family from the perspective of bioinformatics. Therefore, we selected five plants, namely Physcomitrella patens, Selaginella moellendorffii, Picea abies, Triticum aestivum, and Arabidopsis thaliana, to analyze the promoters of their EXPA family genes. The results showed that (Figure6) the promoter of the TaEXPA7-B gene contains the largest number of RHE element (10). In addition, the RHE element is cis-acting element closely related to the growth and development of plant roots. Therefore, we transformed the TaEXPA7-B gene into a model plant of Gramineae, which is more closely related, to further verify its function. We think this figure should not be deleted.

5. Response to comment:The rice experiments seem more coherent... but still I have found some problems. In figure 8A there is no indication of an HA tag in the vector, but in figure 8C there is a western against this tag. Please clarify. Was the tag included in the vector? Are different lines?

Response: In this study, we added an HA tag to the C-terminal of TaEXPA7-B protein, but it was not shown in Figure 8A due to a writing error. Figure 8A has been revised according to the reviewer's suggestion.

6. Response to comment:Also the materials and methods section of these experiments cast a lot of doubts."A HA tag was added to the C-terminus of the TaEXPA7-B gene to detect TaEXPA7-B protein levels. After mixing the roots, stems, and leaves of rice, the total protein was extracted, and Western blot was used to identify the protein expression levels of the TaEXPA7-B gene in the 488 overexpressed rice. The internal reference protein is Actin (2P2) Mouse. " Why authors mix different tissues and do not evaluate expression in each tissue? If the protein expressed by the transgene is not stable in roots this would compromise the interpretation of the results. Also: which antibody have used to detect the HA tag? Please include the information and the manufacturer. I also do not understand how the internal reference is a mouse protein. I assume that is a mistake and authors are referring to the antibody they are using. Please include in figure 8 a western blot from root protein and clarify all this messy information in materials and methods.

Response: There are some problems in identifying over-expressed rice by using mixed samples of roots, stems and leaves of plants, ignoring the differences of gene expression in different tissues. Therefore, according to the reviewer's opinion, we re-identify TaEXPA7-B with only the root tissue as the material and replace the Figure8 (Line 309). In addition, we have supplemented the information about materials and methods in Western blot (Line 147-153).

7. Response to comment:The paper also needs a thorough revision of english. The written is very poor and difficult to understand.

Response: Thank you for your valuable and thoutful comments. We tried our best to improve the manuscript and made some changes to the manuscript. These changes will not influence the content and framework of the paper. And here we did not list the changes but marked in red in the revised paper.

8. Response to comment:Line 62: "rosa rugosa" in italics.

Response: We are extremely grateful reviewer for pointing out this problem. According to the opinions of other reviewers, this sentence has been deleted in the introduction.

Round 2

Reviewer 2 Report

Comments and Suggestions for Authors

Authors significantly improved the manuscript. The RT-PCR experiment presented by Authors is  qualitative in nature.

However, in Fig. 8B and materials and methods Authors should provide the size of tested and reference gene PCR products in bp. If possibile Authors could add as a supplement Fig the original picture of DNA electropherogram with DNA markers, not only the DNA bands, to show their real size.

Also in Fig. 8C and results section provide the molecular weight of fusion protein and actin in kDa. Original pictures of protein electropherograms (stained for example by Coomassie blue) with mol. wt. standards could be very advisable to show the real size of proteins.

Comments on the Quality of English Language

 Minor editing of English language required.

Author Response

  1. Response to comment:However, in Fig. 8B and materials and methods Authors should provide the size of tested and reference gene PCR products in bp. If possibile Authors could add as a supplement Fig the original picture of DNA electropherogram with DNA markers, not only the DNA bands, to show their real size. Also in Fig. 8C and results section provide the molecular weight of fusion protein and actin in kDa. Original pictures of protein electropherograms (stained for example by Coomassie blue) with mol. wt. standards could be very advisable to show the real size of proteins.

Response: According to the requirements of reviewers, the sizes of target gene, reference gene, target protein and reference protein have been indicated in Section 2.4. Moreover, the DNA electrophoresis map with Markers was uploaded. However, we can't provide the complete protein map with Markers at present. Because our samples are all used up, it will take a long time to transform and raise seedlings if we make up the map.(Line 153-155)

Reviewer 3 Report

Comments and Suggestions for Authors

I still have a lot of concerns about the manuscript. Authors argue that the description of how the transgenic plants have been constructed can be found in previous reports, but in this reports, you cannot find the information on many plants used in this work, and how the line was selected.

Also the way of selected the genes is quite diffuse and the answer to my questions is not really convincing. 

Author Response

  1. Response to comment:I still have a lot of concerns about the manuscript. Authors argue that the description of how the transgenic plants have been constructed can be found in previous reports, but in this reports, you cannot find the information on many plants used in this work, and how the line was selected. Also the way of selected the genes is quite diffuse and the answer to my questions is not really convincing.

Response: In this manuscript, all the over-expressed Arabidopsis are the results of previous studies of our group. Our previous research was based on the transcriptome data of winter wheat. In the study, some genes in EXPA family were over-expressed, and it was found that they had a significant role in promoting the growth of plant roots. Therefore, we compared the differences between the roots of Arabidopsis, which was overexpressed in the early stage. In addition, combined with the results of promoter, the TaEXPA7-B gene was screened and transformed into rice to further verify its function.

Round 3

Reviewer 3 Report

Comments and Suggestions for Authors

Now I have clarified the weak points. I can recommend acceptance.